# Determining the Virulence Properties of *Escherichia coli* ST131 Containing Bacteriocin-Encoding Plasmids Using Short- and Long-Read Sequencing and Comparing Them with Those of Other *E. coli* Lineages

**DOI:** 10.3390/microorganisms7110534

**Published:** 2019-11-06

**Authors:** Ana Carolina da Cruz Campos, Francis M. Cavallo, Nathália L. Andrade, Jan Maarten van Dijl, Natacha Couto, Jan Zrimec, Jerome R. Lo Ten Foe, Ana C. P. Rosa, Paulo V. Damasco, Alex W. Friedrich, Monika A. Chlebowicz-Flissikowska, John W. A. Rossen

**Affiliations:** 1Departamento de Microbiologia, Imunologia e Parasitologia, Universidade do Estado do Rio de Janeiro, Rio de Janeiro 20550-170, Brazil; anabio86@gmail.com (A.C.d.C.C.); nathalu84@yahoo.com.br (N.L.A.); anarosa2004@gmail.com (A.C.P.R.); 2Department of Medical Microbiology and Infection Prevention, University Medical Center Groningen, University of Groningen, Hanzeplein 1, 9713GZ Groningen, The Netherlands; f.m.cavallo@rug.nl (F.M.C.); j.m.van.dijl01@umcg.nl (J.M.v.D.); n.monge.gomes.do.couto@umcg.nl (N.C.); j.r.lo.ten.foe@umcg.nl (J.R.L.T.F.); alex.friedrich@umcg.nl (A.W.F.); m.a.chlebowicz@umcg.nl (M.A.C.-F.); 3Department of biology and Biological Engineering, Chalmers University of Technology, Chalmersplatsen 4, 412 96 Göteborg, Sweden; janzrimec@gmail.com; 4Departamento de Doenças Infecciosas e Parasitárias, Universidade Federal do Estado do Rio de Janeiro, Rua Voluntário da Patria, 21, Rio de Janeiro 941-901107, Brazil; paulovieiradamasco@gmail.com

**Keywords:** ST131, *E. coli*, bacteriocins, plasmids, AmpC-beta-lactamase, UTIs, virulence

## Abstract

*Escherichia coli* ST131 is a clinical challenge due to its multidrug resistant profile and successful global spread. They are often associated with complicated infections, particularly urinary tract infections (UTIs). Bacteriocins play an important role to outcompete other microorganisms present in the human gut. Here, we characterized bacteriocin-encoding plasmids found in ST131 isolates of patients suffering from a UTI using both short- and long-read sequencing. Colicins Ia, Ib and E1, and microcin V, were identified among plasmids that also contained resistance and virulence genes. To investigate if the potential transmission range of the colicin E1 plasmid is influenced by the presence of a resistance gene, we constructed a strain containing a plasmid which had both the colicin E1 and *bla_CMY-2_* genes. No difference in transmission range was found between transformant and wild-type strains. However, a statistically significantly difference was found in adhesion and invasion ability. Bacteriocin-producing isolates from both ST131 and non-ST131 lineages were able to inhibit the growth of other *E. coli* isolates, including other ST131. In summary, plasmids harboring bacteriocins give additional advantages for highly virulent and resistant ST131 isolates, improving the ability of these isolates to compete with other microbiota for a niche and thereby increasing the risk of infection.

## 1. Introduction

Pathogenic Extraintestinal *Escherichia coli* (ExPEC), including the uropathogenic *E. coli* (UPEC) pathotype, is often associated with urinary tract infections (UTIs) [1]. They carry a high number of virulence factors such as adhesins, fimbriae, hemolysins, aerobactin and others that allow these bacteria to live in the human gut but also, to cause infections at other sites [2,3,4]. One of the survival strategies of *E. coli* is the production of bacteriocins, a group of antibacterial peptides often encoded by genes located on plasmids and able to kill normally closely related surrounding bacteria [5]. Although not required for growth, they help to outcompete other microorganisms (bacteria and fungi) for the limiting nutrients in the environment [6,7]. Colicins and microcins are the types of bacteriocins most often found in pathogenic and in approximately 30% of commensal *E. coli* [8,9,10]. Colicin genes are mostly located in operons, also containing the colicin immunity gene, important for neutralizing its toxic effect on the producer strain, and the lysis gene required for colicin release. The operon is activated by the SOS system [11]. In contrast, microcins are not inducible by the SOS system and are not toxic to producer strains. Colicins have different ways of action. They can act by forming a pore in the bacterial membrane, digesting bacterial DNA by their nuclease activity, or by interfering with cell wall synthesis [5,12]. The presence of multiple bacteriocins in *E. coli* isolates is common and increases their urovirulence [13] and the development of bacteremia of urinary tract origin [9]. 

Within the *E. coli* population, ST131 is one of the most successful lineages frequently causing UTIs and bloodstream infections (BSIs). Their success is partially associated with the presence of fluoroquinolone resistance (FQR), β-lactamases responsible for their ESBL phenotype and specific virulence genes [14,15]. Among a diversity of beta-lactamases genes, *bla_CMY-2_* is frequently identified. It encodes for an AmpC type of β-lactamase that confers resistance to all β-lactam antibiotics except the fourth-generation cephalosporins and carbapenems [16]. ST131 *E. coli*, frequently belonging to phylogenetic group B2, are associated with a high virulence profile [17]. Due to this combination of a high resistance and virulence profile, infections caused by the ExPEC ST131 are a serious threat for patients. In addition, other successful and worldwide spread lineages, such as ST405 and ST648, have been associated with antibiotic resistance [18]. As mentioned, the production of bacteriocins, such as microcin, by ExPEC is associated with a more virulent profile of the bacterium [2]. Whole genome sequencing (WGS) has been used to reveal the epidemiology and evolution of the ST131 lineage [19,20] and the role of mobile genetic elements (MGE) herein [21]. MGEs, particularly plasmids, can be easily exchanged between isolates creating sub-lineage variants that become even more resistant and virulent [22]. As mentioned, in *E. coli*, several bacteriocins are encoded by genes located on plasmids and the presence of resistance genes and specific bacteriocins on single plasmids could potentially contribute to the success of ST131 *E. coli*. However, to the best of our knowledge, there are no studies that address the importance of plasmids encoding both resistance and bacteriocin genes for the successful dissemination and virulence potential of this high-risk clones and only a few studies have investigated the importance of bacteriocins with respect to the virulence of these successful lineages. Therefore, our study aimed to investigate the bacteriocin in vivo activity in *E. coli* isolates, focusing on successful lineages, particularly ST131, and to characterize resistance genes and bacteriocin-encoding plasmids of ST131 *E. coli* isolated from clinical urine samples in an attempt to reveal their role in the bacterial virulence. In addition, we also studied the association between bacteriocins and the phylogenetic groups, and resistance and virulence profiles. 

## 2. Materials and Methods

### 2.1. Bacterial Isolates and Bacteriocin Activity Assay

The isolates used in this study were obtained from urine samples of patients hospitalized in four hospitals located in Rio de Janeiro, Brazil. *E. coli* isolates were identified using mass spectrometry (Bruker, Bremen, Germany). Antibiotic susceptibility testing was performed using VITEK-2 (bioMérieux, Marcy l’Etoile, France), for which isolates were cultured on cysteine lactose deficient medium agar plates (CLED, BD, Heidelberg, Germany) until a cell density higher than 10^5^ colony-forming units (CFU) was obtained. Bacterial cells were stored at −80 °C in a Lysogeny Broth (LB, Merck, S.A.) with 20% glycerol before being further analyzed. For this study, isolates were selected that belong to ST10, ST69, ST73, ST131, ST475 and ST648 as they are the most prevalent STs in the Rio de Janeiro population. In addition, isolates containing a bacteriocin gene were selected for further analyses. For the bacteriocin activity assay, the isolates were grown in 5 mL of LB (Luria Both) overnight at 37 °C. Subsequently, 5 μL of the bacterial suspension was added to LB agar plates containing *E. coli* isolates or the control strain (*E. coli* K-12 MG1655) and plates were incubated overnight at 37 °C. Formation of an inhibition zone (halo) around the producer indicated production of a bacteriocin by the isolate in the suspension and sensitivity to it of the isolate already on the plate. 

### 2.2. Short-Read Sequencing

DNA of the isolates was extracted using the UltraClean microbial DNA isolation kit (MO Bio laboratories, Carlsbad, CA, USA) and library preparation was performed using the Nextera XT kit (Illumina, San Diego, CA, USA) and the library was sequenced on a Miseq (Illumina), as described previously [23]. The raw data were deposited in the European Nucleotide Archive under the project number PRJEB23420.

### 2.3. Long-Read Sequencing

For the bacteriocin-producing ST131 isolates 5332, 5848 and 7078, whole genome sequencing (WGS) was also performed using Oxford Nanopore Technologies (ONT, Oxford, UK) long-read sequencing. DNA extraction was performed as mentioned above for the short-read sequencing. For this, several library preparation kits were used: isolate 5848 was prepared using the 2D ligation sequencing kit (SQK-LSDK208), isolates 5332 and 7078 were prepared with the Rapid Sequencing kit (SQK-RAD004) according to the manufacturer instructions. The libraries were loaded onto two different flo-MIN106 R9.4 flow cells. The runs were performed on a MinION device (ONT). Base calling was performed using Albacore v1.2.2 (ONT) with the r94_250bps_2d.cfg workflow for isolate 5848 and with Guppy v3.2.2 (ONT) for the other two isolates. 

### 2.4. Assembly, Annotation and Analysis

For the short-reads, de novo assembly was performed with CLC Genomics Workbench v12.0 (Qiagen, CLC bio A/S, Aarhus, Denmark) using the default settings and an optimal word-size. Annotation was performed by uploading the assembled genomes onto the RAST server version 2.0 [24]. The ST and virulence genes were identified by uploading the assembled genomes in fasta format to the Center for Genomic Epidemiology (CGE), MLST finder website (version 1.7) [25] and VirulenceFinder (version 2.0) [26]. For the long-reads, we analyzed the quality of the data through Poretools v0.6.0 [27] and transformed the fast5 files into fastq files using the same tool [27]. Subsequently, we performed hybrid assemblies using Illumina short reads and ONT long reads using Unicycler v0.4.1[28]. To visualize the assembly graphics, we used Bandage v0.8.1 [29]. 

### 2.5. Plasmid Analysis and Identification of Bacteriocin Genes

The plasmids incompatibility groups were identified by uploading the assembled files, generated using the hybrid assembly approach described above, to PlasmidFinder (version 1.3) [26]. The plasmid sequences were annotated automatically using the RAST server version 2.0 and manually using CLC Genomics Workbench v12.0 (Qiagen, CLC bio A/S, Aarhus, Denmark). Subsequently, plasmids were uploaded to BLAST (NCBI database) to identify the closest reference plasmids. Alignment of plasmid sequences was done using Easyfig v2.2.3 [30] and DNA plotter [31]. The bacteriocin genes present in the isolates were detected by BAGEL 3, by uploading the fasta files onto the online tool [32]. Plasmid mobility was analyzed by locating and typing the origin-of-transfer (oriT) regions using a DNA structural alignment algorithm that finds minimal Euclidean distances and *p*-values between query oriTs and target plasmids [33]. Potential transfer host ranges of the predicted MOB groups were determined from a MOB-typed dataset [34].

### 2.6. Bacterial Transformation Assay

Isolate 5848 (plasmid donor) was cultivated in LB supplemented with 100 µg/mL of cefotaxime overnight at 37 °C. Then, plasmid extraction was performed using the innuPREP plasmid Mini Kit (Analytikjena Jena, Germany) according to the manufacturer’s protocol. The plasmid was identified by size selection on an agarose gel and isolated from it using the innuPrep gel extraction kit (Analytikjena, Germany) according to the manufacturer’s protocol. After purification, the plasmid DNA was quantified and used to transform *E. coli* Dh5α using CaCl_2_ and a heat shock, as previously described [35]. The bacteria were plated onto LB agar plates containing cefotaxime (1 mg/L) (Mediaproducts, Groningen, The Netherlands) and resistant colonies were considered to have successfully acquired the plasmids. They were tested for the presence of the plasmid using gel electrophoreses and further submitted to short- and long-read sequencing, as described above. The plasmid sequences from the recipient were compared with the sequence of the plasmid identified in donor sample 5848. Annotation of the plasmids’ hybrid assemblies was performed using RAST (version 2.0) [24] and manual annotation in CLC Main Workbench.v11.0.1 (Qiagen, CLC bio A/S/Aarhus, Vedbæk, Denmark). 

### 2.7. Competition Adhesion and Invasion Experiments

Bacteria-cell adhesion assays were performed using Human Embryonic Kidney (HEK293)cells (ATCC^®^LGC), maintained in DMEM medium (ThermoFisher, Paisley, Scotland) containing 2% of fetal bovine serum (VWR, Roden, The Netherlands). Cells were grown to form a monolayer in 24-well plates and inoculated with different strains in triplicate. After incubation at 37 °C in a CO_2_ incubator for 3 h, the cells were washed three times with phosphate-buffered saline (PBS) and then lysed using PBS containing 0.1% Triton X-100. The lysates were collected, serially diluted and the dilutions were plated onto TSB agar plates. Adhesion was calculated as the number of CFU/mL recovered per well. For invasion, the cells were incubated for an additional 1.5 h with medium containing 100 µg/mL kanamycin. After incubation, cells were lysed as described above, serially diluted and plated on LB plates and the number of CFU was determined. 

### 2.8. Competition Fitness Assay

To investigate if the presence of antibiotic resistance and colicin-encoding plasmids affect the bacterial fitness, a relative fitness test was performed, as described previously [36] with some minor modifications. Briefly, relative fitness was estimated by a pairwise competition assay for which isolates were grown in M9 minimal medium for 24 h. Subsequently, a single colony was used to inoculated M9 minimal medium for overnight growth at 37 °C. The initial cell density was measured by plating different dilutions on both antibiotic-free LB agar and on LB agar supplemented with 1 mg cefotaxime (A0 and B0). The next day, cultures of the bacteria were serially diluted in 0.9% NaCl in a 96-well plate. Subsequently, the different dilutions were also plated on both an antibiotic-free LB agar and on LB agar supplemented with 1 mg cefotaxime and incubated for 24 at 37 °C (A1 and B1). The relative fitness (W) was calculated as the ratio of the Malthusian parameter of each competitor: WAB = MA/MB, where MA = ln[A(1)/A(0)], MB = ln[B(1)/B(0)], A(0) and B(0) the estimated initial densities of A and B and A(1) and B(1) the estimated densities of A and B after 24h. This experiment was repeated 8 times.

### 2.9. Statistical Analysis

For the adhesion, invasion and biological fitness assays, the results obtained for the isolates were analyzed thought the Student’s t-test using GraphPad Prism v.7.04. The association between the presence of bacteriocin genes with phylogenetic groups, virulence genes and antibiotic multidrug resistance profiles were performed by the Fisher’s exact test using the GraphPad Prisma v.7.04. *p*-values < 0.05 were considered as statistically significant (GraphPad Software, La Jolla, USA). 

### 2.10. Ethical Considerations

This study was submitted and approved on October 2015 by the Pedro Ernesto University Hospital ethical committee 174 according to Brazilian legislation and received the following registration number: CAAE number: 45780215.8.0000.5259. All the samples used in this study were obtained from patients that signed a consent form in which the gave permission for the use of sample and data for this study.

## 3. Results

### 3.1. Bacterial Isolates and Bacteriocin Sensitivity Profile

A collection consisting of 69 *E. coli* isolates obtained from urine samples of hospitalized patients in Brazil were included in this study, of which 41 belonged to phylogenetic group B2, 13 to phylogenetic group D, 9 to phylogenetic group A, 5 to phylogenetic group B1 and 1 to phylogenetic group F. The distribution of the phylogenetic groups over the different ST-types is indicated in Table 1. Three bacteriocin-producing ST131 isolates were identified: fim*H*22-O25:H4 isolates 5332 and 5848 and fim*H*30-O25:H4 isolate 7078. All three inhibited the growth of isolates belonging to other STs, as well as isolates belonging to ST131 (Figure 1). The rates of sensitivity against bacteriocin-producing ST131 isolates per ST types were: 15.3% (*n* = 4) of ST131, 11.1% (*n* = 1) of ST69, 71.4% (*n* = 5) of ST10, 50% (*n* = 2) of ST73, 25% (*n* = 1) of ST405 and 66.6% (*n* = 4) of ST648. Among singleton ST types, two isolates were sensitive to the bacteriocin(s) produced by the three bacteriocin-producing ST131 isolates. Whereas, 6419 was only sensitive to the bacteriocin produced by the *H*30-ST131 (7078) isolate, 7167 was only sensitive to bacteriocins produced by the *H*22-ST131 (5332 and 5848) isolates. Among the other ST types, our analysis revealed that isolates 6419 (ST414), 6492 (ST12), 9097 (ST95) and 9307 (ST91) belonging to phylogenetic group B2; isolates 3921 (ST10), 5038 (ST58), 5306 (ST641), 7167 (ST1431) and 6632D (ST453) belonging to phylogenetic group B1; and isolates 1825 (ST93) and 7500 (ST744) belonging to phylogenetic group A were able to inhibit the growth of isolates from different ST types (Figure 2). In particular, isolates 3921, 9097 and 9307 were able to inhibit the growth of the majority of ST131 isolates. The rates of sensitivity against bacteriocin produced by non-ST131 isolates were 76.9% (*n* = 20), 83.3% (*n* = 5), 55.5% (*n* = 5), 57.1% (*n* = 4), 50% (*n* = 2), 25% (*n* = 1) and 61.5% (*n* = 8) for ST131, ST648, ST69, ST10, ST73, ST405 and singleton ST types, respectively. 

### 3.2. Distribution of Bacteriocin Genes among Clinical Isolates

Analyses of WGS data revealed the presence of bacteriocin genes encoding microcin V, pyocin S, and the colicins A, Ia, Ib, M and E1 in our bacteriocin-producing isolates (Table 2). In the ST131 isolates showing bacteriocin activity, i.e., 5332, 5848 and 7078, the colicin E1, Ia, Ib and the pyocin S genes were present. Moreover, isolates 5848 and 7078 also contained a microcin V gene. ST131 isolate 7104 presented the colicin E1 and pyocin S genes, whereas all of the remaining ST131 isolates contained only the pyocin S gene. In all ST73 isolates, the colicin E9 gene was identified and an additional bacteriocin was identified in isolates 7348 (colicin-10), 2723A (colicin E1) and 9492 (pyocin S). ST10 isolates also contained genes for different bacteriocins, such as a microcin B17 in isolate 6077, microcin V, colicin A and colicin M in isolate 8874, colicin A and M in isolate 5217 isolate, and colicin E1 in isolate 8200. Bacteriocin genes found in bacteriocin-producer isolates are listed in the Appendix A. No known bacteriocin genes were detected in ST69, ST405 and ST648 isolates.

### 3.3. Association between Phylogenetic Group, Virulence Genes, MDR and Bacteriocins Genes

Our analysis revealed an association between phylogenetic group B2 and the presence of bacteriocin genes (*p* = 0.0219). We also analyzed a possible correlation between the presence of bacteriocins and 68 different virulence genes and found a statistically significant association between the presence of bacteriocins genes and the virulence genes *iroN* (*p* = 0.006), *fyuA* (*p* = 0.026), *fhuA* (*p* = 0.034), *irp2* (*p* = 0.026), *nleA* (*p* = 0.012), *sigA* (*p* > 0.001) and *ompT* (*p* = 0.046). No statistically significantly association was found between the MDR profile and the presence of bacteriocin genes.

### 3.4. Identification and Characterization of Plasmids Present in ST131 Isolates

To determine if identified colicin genes were located on the bacterial chromosome or on a plasmid, a combination of short- and long-read sequencing of three ST131 isolates (5332, 5848 and 7078) was used. Hybrid assemblies revealed the presence of three different plasmids in isolates 5332 and 5848, of which two were also identified in isolate 7078 (Figure 3 and Appendix A). The smallest plasmid (6645 bp), designated here as p5848A1, belonging to replicon type Col156, contained the colicin E1 gene operon, consisting of the colicin E1 gene and the genes encoding for its immunity and lysis proteins, but no resistance genes. Other genes present on this plasmid were the *mebA*, *mebB*, *mebC*, *mebD* genes, two genes encoding the entry exclusion proteins 1 and 2, and two genes encoding hypothetical proteins (Figure 3A). The medium-size plasmid (101,573 bp), designated in this study as p5848A2, belonged to incompatibility group IncI1. It mostly contained genes encoding for hypothetical or mobile element proteins, but also the beta-lactamase gene *bla_CMY-2_* flanked by the Colicin Ib and the *blc* genes, the latter encoding for a membrane-associated lipoprotein. Other genes located on this plasmid were the *sugE* gene, conferring resistance to quaternary ammonium compounds, *tra* genes known to be involved in conjugation, the antitoxin genes *phD* and *ccdA*, the toxins genes *doC* and *ccdB*, the *umuC* gene related with activation of the SOS system, and the *psiB* and *psiA* genes involved in inhibition of the SOS system. In addition, transposases belonging to the IS200/IS605 family were detected in this plasmid. This plasmid is very similar to the pSTM709 plasmid present in *Salmonella enterica* subspecies *Typhimurium* (NCBI reference sequence: NC_023915.1) isolated from Uruguay, except for three regions, one region (nt 2-408) absent in the p5848A2 plasmid and two regions (nt 38446-41078 and nt 101808-101573) only present in p5848A2 (Figure 3B and Appendix A). Finally, the largest plasmid (149,732bp), designated here as p5848A3, belonged to incompatibility groups IncFII and IncFIB and carried several genes, most of them specifying hypothetical proteins and proteins associated with plasmid conjugation, transcriptional regulation, and mobile genetic elements. It also contained virulence genes, including genes for iron uptake, such as the aerobactin genes *iucAD*, the ABC iron transporter genes *iroBN*, the hemolysin gene *hha*, genes for the bacteriocins colicin Ia and microcin V. Further, this plasmid carried antibiotic resistance genes, including tetracycline resistance genes (*tetA* and *tetR*) and the trimethoprim resistance *dfrA5* gene. Other genes present on this plasmid encode the Macrolide-specific ABC-type efflux system (*macA* and *macB*) and ion transporters (*copB*, *merC*, *merE* and *merT*) (Figure 3C).

### 3.5. Bacterial Fitness Cost and Predicted Transfer Potential

We decided to investigate the potential effect of bacteriocins on the maintenance and transmission of plasmids that also contain antibiotic resistance genes. We constructed a transformant strain Dh5α +, containing a plasmid that combines the *bla_CMY-2_* flanked by a gene encoding a hypothetical protein and the *blc* and *sugE* genes, recombined with the ColE1 plasmid p5848A1, here designated p5848A1.2 (accession number: PRJEB34226) (Appendix A). First, we assessed the fitness cost of carrying bacteriocin- and resistance gene-encoding plasmids, and calculated the relative fitness using the ratio of the Malthusian parameter. The relative fitness of DH5α+ ranged from 0.211–0.95 and no statistically significant difference in the growth rate was observed between the control sample DH5α and the transformant strain DH5α+ (Figure 4). We predicted plasmid mobility using the oriT regions by MOB typing [36]. Our results showed that plasmid p5848A1.2 belongs to MOB group P, similar to the ColE1 plasmid p5848A1. Therefore, there is no indication for a difference in transfer range among the plasmid containing a bacteriocin and the one containing both a bacteriocin and a cephalosporin resistance gene. In addition, plasmid p58548A2 that also contains genes for colicin Ib and cephalosporin resistance, belongs to the same MOB group P, while plasmid p5848A3 that encodes colicin Ia, microcin V and other resistance genes belongs to the MOB group F (Table 3).

### 3.6. Bacteriocin-Encoding Plasmids and Phenotypical Virulence Profile

We investigated if the presence of plasmids affects the virulence profile of isolates through testing the ability to adhere and invade urinary tract epithelial cells. As expected, our results showed that the bacteriocin-producing isolates 5848, 5332 and the DH5α+ (transformant) had a higher adherence to HEK-293 cells, compared to the negative control (DH5α). However, from the non-bacteriocin producing isolates only isolate 1643 showed a statistically significant lower adherence compared to isolate 5332 (*p* = 0.0429) and no other statistically significant differences were observed. Similar results were found for the invasion ability where no differences were found for colicin-producing isolates compared to non-colicin-producing ones. However, the DH5α+ (transformant) strain presented a significant increase in the ability to adhere and invade the urothelial cells (*p* = 0.0001) compared to the control DH5α strain, presenting similar results as found for isolates 5332 and 5848 (Figure 5A,B).

## 4. Discussion

Bacteriocins are mediators of intra- and interspecies interactions. Whereas recent studies have focused on bacteriocins as a potential replacement for antibiotic therapy, their potential as a virulence factor for specific successful *E. coli* lineages remains poorly investigated [9,37,38]. In the present study, we investigated the bacteriocin activity of clinical *E. coli* isolates obtained from urine samples of patients hospitalized in Brazil, with a focus on ST131. Three clinical ST131 *E. coli* isolates contained plasmids with bacteriocin genes (colicins E1, Ib, Ia, and microcin V). Two of the isolates belonged to the fim*H*22 and serotype O25:H4, while another isolate was identified as fim*H*30 and O25:H4. The *H*22-ST131 sublineage is frequently isolated from UTIs and, because it was also detected in poultry, it is considered as a foodborne uropathogen [39]. The *H*22-ST131 isolates from poultry and the isolates in this study shared the presence of a bacteriocin-containing plasmid. Interestingly, the *H*30-ST131 *E. coli* isolate containing colicin E1 and Ib genes located on plasmids, was sensitive for fluoroquinolones and cephalosporins, whereas both *H*22-ST131 isolates were AmpC-β-lactamase-producing and one of them (5332) was fluoroquinolone-resistant. Horizontal transfer of the plasmid between isolates within the same geographic area could have occurred. However, also many ST131 isolates were circulating in the same area without these plasmids. Therefore, the presence of similar plasmids in both resistant *H*22-ST131 and susceptible *H*30-ST131 isolates could also be explained by the fact that *H*30 strains evolved from the *H*22 lineage [21].

In our study, the presence of plasmids carrying colicin Ia, Ib and E1, and microcin V genes, in the ST131 isolates, was the most likely cause of their ability to inhibit the growth of other *E. coli* isolates. Our results show that colicin produced by ST131 isolates was able to kill other ST131 isolates resulting in strain–strain competition. This result agrees with findings in previous studies showing that colicins act mainly against closely related *E. coli* isolates [12,38]. Our results also show that a higher number of the investigated ST131 isolates is sensitive to the bacteriocin(s) produced by non-ST131 isolates than for bacteriocin(s) produced by the ST131 isolates. This is in contrast with previous studies that reported bacteriocin insensitivity of isolates belonging to phylogenetic group B2, to which ST131 also belongs [40]. However, another study pointed out that this insensitivity to bacteriocins of ST131 may have been overestimated and is related to the O-antigen [41].

Plasmid p5848A1 containing the colicin E1 gene has a similar structure as other colicin E1 containing plasmids, including the presence of the *meb* operon, which plays a role in conjugation similar to the *traY* genes of IncF plasmids [42]. The plasmid p5848A2, containing colicin Ib, was found to be highly similar to a plasmid (pSTM709) present in *Salmonella enterica* subspecies *Typhimurium* isolated in Uruguay, but had two additional regions encoding mobile elements, hypothetical proteins, and the RepY and InC proteins, that are absent from the pSTM709 plasmid. The presence of colicin Ib conferred a competitive advantage in the gut to *Salmonella enterica* [10]. The presence of genes encoding the membrane lipoprotein *blc* and the efflux pump *sugE* flanking the beta-lactamase *bla_CMY-2_* gene, is similar to the structure identified for other plasmids among pAmpC-producing *E. coli* from poultry and fecal human samples [43]. The largest plasmid p5848A3, harboring both resistance and virulence genes, carries the colicin Ia and microcin V genes. Notably, colicin Ia is induced by the SOS system, while the microcin V is induced by iron-limited conditions. The same plasmid also carries several genes for iron uptake, consistent with a previously described association between siderophores and salmochelin [13]. The combination of these genes on the same plasmid may give an advantage to survive in and to colonize iron-limited environments as the urinary tract. Furthermore, the presence of colicin Ia and microcin V genes on the same plasmid was described in another study as a non-aleatory evolution that may have happened to increase the killing range and/or to increase the fitness advantage of bacteria in different environments [44].

Eleven non-ST131 isolates were identified, which have the ability to inhibit the growth of isolates from different ST types. The bacteriocin genes among ST131 isolates were less diverse than those in non-ST131 isolates, which is in agreement with previous findings [2]. Interestingly, despite the resistance of ST69 and ST405 to bacteriocins, we did not identify any bacteriocin genes nor did we find any other known genes related to bacteriocin resistance in these groups. Therefore, it would be highly interesting to investigate the mechanism(s) behind the observed resistance in future research. Furthermore, we identified a statistically significant association between the presence of bacteriocin genes and the phylogenetic group B2 among our isolates. These results are different from previous findings showing such an association also for isolates belonging to phylogenetic group D [2,45]. However, our phylogenetic group D isolates did not contain any known bacteriocin genes.

In our study, colicin E1 genes were identified in both bacteriocin-producing and non-producing ST131 isolates. This colicin has already been described to be a potential virulence factor for uropathogenic *E. coli* strains and is frequently found in UPEC [46]. The colicin Ib gene was identified in all ST131 bacteriocin-producing isolates in our study and was found before to play a role in the competition with intestinal *E. coli* [10]. Together with the colicin Ia and microcin V, these bacteriocins are highly frequent in successful clones causing symptomatic and asymptomatic UTIs [47]. The presence of bacteriocin-encoding plasmids may increase the survival and help compete against other *E. coli* in the gut, resulting in colonization of the gut by these sublineages, thereby increasing their risk to cause UTIs, particularly, because UTIs can start with the contamination of the periurethral region by bacteria present in the gut [48]. Such association between high colonization rates and UTIs was already described for the beta-lactamase-producing O25:H4 ST131 clone [47]. In addition, the presence of bacteriocin(s) in high-risk clones can lead to long-term colonization and persistence of ESBL-producing ST131 *E. coli* [49,50].

Previous studies reported the importance of plasmids in the evolution of sublineages, particularly the highly resistant and virulent *H*30Rx-ST131 [21]. The presence of both bacteriocin- and resistance gene carrying plasmids in *H*22 and *H*30 ST131 isolates may increase the chance of spread of those plasmids among the more antibiotic resistant sublineages as *H*30-R and *H*30Rx-ST131 or less resistant *H*41 and enhance the chances for the emergence of other high-risk sublineages. It has been shown that the evolution of successful lineages such as ST131 is driven by the efficiency of obtaining a high-resistance and -virulence profile with less metabolic stress, i.e., with lower fitness cost. We identified a positive association between the presence of some virulence and bacteriocin genes. This result is consistent with previous studies that showed an association between the presence of common virulence genes and bacteriocin genes in UPEC [9,13]. Most of the virulence genes associated with bacteriocins in our study were iron uptake genes, supporting the view that the *fyuA* and *iroN* genes are associated with different bacteriocins [51]. In addition, we did not identify a positive association between the MDR profile and the presence of bacteriocin genes, which is in agreement with the results of a previous study [9].

Through phenotypical analysis, we investigated whether the presence of plasmids encoding resistance, bacteriocins and other virulence factors affects the adhesion and invasion ability to uroepithelial cells, which are key events in UTIs’ pathogenesis [48,52]. Our results show that the presence of three bacteriocin-encoding plasmids seems not to affect the adhesion and invasion ability, since no significant difference was found between the bacteriocin-producing isolates and most of the non-bacteriocin producing isolates. Further, we investigated whether the presence of plasmids containing resistance and bacteriocin genes could improve the virulence of the bacteria without affecting the spread and maintenance of this plasmid. For this, we constructed a DH5α strain (DH5α +), containing a ColE1 plasmid (p5848A1.2), carrying both the *bla_CMY-2_* and colicin E1 genes. This strain showed a significant increase in the ability to adhere and invade urothelial cells compared to the parental DH5α strain. Also, a previous study showed that the carriage of ESBL-encoding plasmids improves the virulence in some strains [53]. In addition, a higher copy number of the plasmid in the transformant strain compared to that of the plasmid in clinical isolates, and thus a higher expression of the bacteriocin gene located on it, may be associated with an increase in virulence. However, no difference in virulence was observed between the transformant strain and the clinical isolates. Moreover, the presence of this plasmid does not seem to increase the overall bacterial fitness cost. Furthermore, the original plasmid containing the colicin E1 (p5848A1) and the modified plasmid containing the colicin E1 and *bla_CMY-2_* genes belong to the same MOB subtype and, therefore, are expected to have the same transfer range. This means that the acquisition of a resistance gene seems not to affect the structural properties of the transfer range. These results are in agreement with previous findings showing that ColE1 plasmids are important vehicles for antibiotic resistance and other traits in Enterobacteriaceae [42]. Together, the presence of ESBL and bacteriocin plasmids in *E. coli* ST131 isolates could improve the bacterial competition ability and resistance without an increase in the fitness cost, which would make the transmission of these plasmids more likely.

## 5. Conclusions

In summary, the combined presence of resistance genes, such as the beta-lactamase *bla_CMY-2_* gene, and bacteriocin-containing plasmids in highly resistant and virulent *E. coli* ST131 high-risk isolates is a concern, as it may result in increased colonization rates, thereby enhancing the chances for causing UTIs. The fact that these plasmids do not reduce the adherence and invasion potential and do not come with a high fitness cost, increases the chance of spread, including spread to other bacterial species and to already difficult-to-treat sublineages of ST131, such as *H*30Rx-ST131. Despite the fact that the virulence potential of isolates carrying these plasmids still needs to be further evaluated in vivo, their identification and characterization using molecular approaches are undoubtedly important to monitor the evolution of these high-risk clones.

## Figures and Tables

**Figure 1 microorganisms-07-00534-f001:**
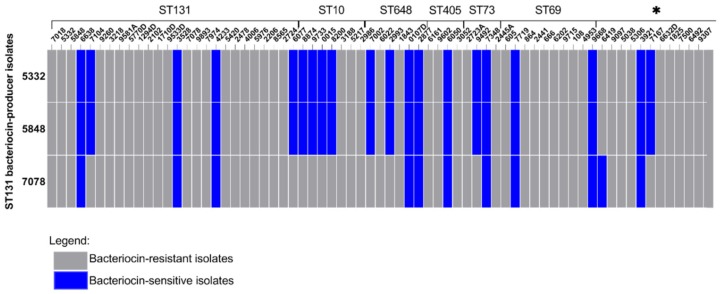
Bacteriocin activity among ST131. The graph shows the bacteriocin activity of the three bacteriocin-producer ST131 isolates against other isolates that belong to ST131, ST10, ST648, ST405, ST73, ST69, ST297 and ST414. The * indicates the singleton isolates. The dark blue hits indicate the isolates that were sensitive to bacteriocin and the grey hits indicate bacteriocin-resistant isolates.

**Figure 2 microorganisms-07-00534-f002:**
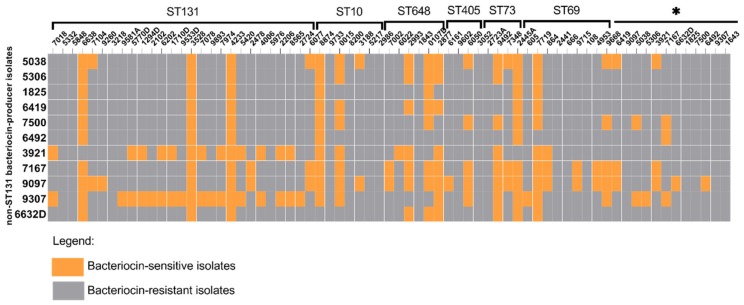
Bacteriocin activity among non-ST131 isolates. The graph shows the bacteriocin activity of non-ST131 isolates against isolates from different ST types. The * indicates the singleton isolates. Orange represents isolates that were sensitive to the bacteriocin produced by the non-ST131 isolates and grey represents the bacteriocin-resistant isolates.

**Figure 3 microorganisms-07-00534-f003:**
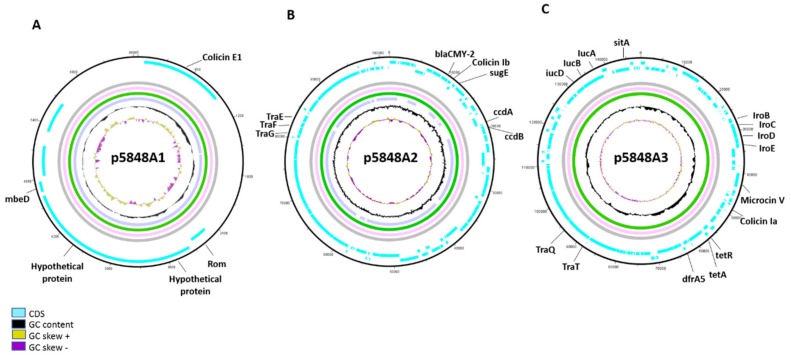
ST131 plasmid sequence alignments using hybrid assemblies. (**A**) Sequence alignment of plasmid p5848A1 found in isolates 5332, 5848 and 7078 containing the colicin E1 gene. (**B**) Alignment of plasmid p5848A2 found in isolates 7078, 5332 and 5848, containing the beta-lactamase *bla_CMY-2_*, the *sugE* and the colicin *Ib* genes. (**C**) Alignment of plasmid p5848A3 found in isolates 5332 and 5848, containing the colicin Ia and microcin V genes, the resistance genes *dfrA5*, *tetA* and *tetR* and the virulence genes *iucA, sitA, iroB, iroC, iroD* and *iroE.* Isolate 5332 is indicated in pink, 5848 in green and 7078 in light purple. In addition, blue indicates the coding sequence region (CDS), black indicates the GC content, yellow indicates regions with a positive GC skew, and dark purple yellow indicates regions with a negative GC skew.

**Figure 4 microorganisms-07-00534-f004:**
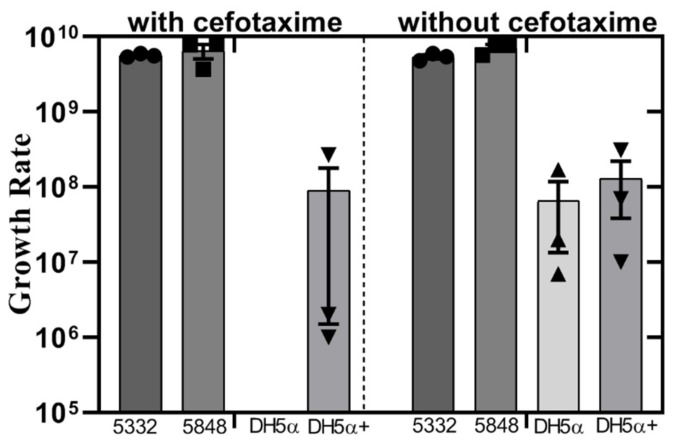
Growth rate of isolates with or without bacteriocin and resistance genes containing plasmids in the presence or absence of cefotaxime. Results showing the growth rate (UFC/mL) of each competitor. Squares, triangles and circles indicate replicates. No statistically significant difference was found when comparing growth rates of DH5α (wild type) and DH5α+ (transformant strain).

**Figure 5 microorganisms-07-00534-f005:**
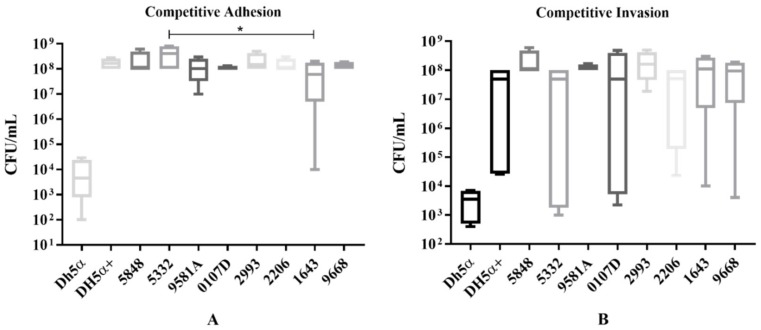
Competitive adhesion and invasion assay. (**A**) Adhesion to HEK-293 (human embryonic kidney cell line) calculated in CFU/mL. (**B**) Invasion of HEK-293 for the same isolates. In both assays, results obtained from the plasmid containing isolates 5332 and 5848 were compared with other ST131 clinical isolates (9581A, 2206, 0107D), with ST648 (2993), ST354 (1643), ST297 (9668) and ST69 (7719) isolates, and with both the parental DH5α and the transformant DH5α+ strains. The * indicates a statistically significant difference (*p* ≤ 0.05).

**Table 1 microorganisms-07-00534-t001:** Bacterial isolates, their ST types, serotypes and phylogenetic groups.

Isolates ID	ST Type	N^o^ of Isolates	Serotype	Phylogenetic Group
3218, 5332, 7018, 7104, 9260,3218,9581A, 5770D,6638, 1294D, 5848, 2102, 1710D, 9533D, 3528, 7078, 9893, 7974, 4233, 5420, 4006, 5976, 2206, 8565, 2724, 6202	ST131	26	O25:H4	B2
2445A,7719,864,2441,666,9715,108,4953,605	ST69	9	O17/O44:H18, O17/O77:H18, O15:H18, O15:H2, O25:H18, O45:H45	D
0015, 6077, 9733D, 5217, 8874, 3188B, 8200	ST10	7	O107:H54, O9:H9, O128ab:H10, O9:H12, O89:H10, O12:H4	A
1843,0107D, 6022, 2986,2993, 7002	ST648	6	O1:H6	B2
6050, 9602, 6161, 2877	ST405	4	O102:H6	D
3052, 9492, 7348, 2723A	ST73	4	O6:H1, O22:H1	B2
9668	ST297	1	O86:H49	B2
9097	ST95	1	O50/O2:H7	B2
6419	ST414	1	O16:H5	B2
5038	ST58	1	O8:H25	B1
5306	ST641	1	O30:H25	B1
3921	ST101	1	O21:H21	B1
7167	ST1431	1	O8:H19	B1
6632D	ST453	1	O23:H16	B1
1825	ST93	1	O7:H4	A
7500	ST744	1	O89:H10	A
6492	ST12	1	O4:H5	B2
9307	ST91	1	O39:H4	B2
1643	ST354	1	O25:H34	F

**Table 2 microorganisms-07-00534-t002:** Distribution of bacteriocin genes among clinical isolates.

Bacteriocin Groups	Bacteriocin Genes	Activity	Bacteriocin Producer Isolates (*n*)	Non-Bacteriocin Producer Isolates (*n*)
B	Ia	Pore-forming	13	0
	Ib	Pore-forming	1	0
	M	Peptidoglycan Synthesis inhibitor	3	2
	10	Pore-forming	0	1
A	E1	Pore-forming	3	7
	E9	DNase	2	4
	A	Pore-forming	1	2
	Pyocin S	DNase	7	24
	Microcin V	Membrane disruption	4	1
	Microcin B17	Membrane disruption	1	1

**Table 3 microorganisms-07-00534-t003:** Results of the prediction of transfer range using origin-of-transfer.

Plasmid ID	Nic Location	Orientation	MOB Subgroup	NIC	*p*-Value
P5848A1	2350	RC	P	−92.6330	<10^−16^
P5848A1.2	8823	F	P	−56.5174	<10^−16^
P5848A2	45897	RC	P	−377522	<10^−16^
P5848A3	110908	F	F	−48.8684	<10^−16^

MOB: mobility groups; nic: relaxase enzyme nicking sites (region within the origin transfer). F, forward; RC, reverse complement.

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
