# Peer review of "Determining the Virulence Properties of Escherichia coli ST131 Containing Bacteriocin-Encoding Plasmids Using Short- and Long-Read Sequencing and Comparing Them with Those of Other E. coli Lineages"

_microorganisms, 2019, doi:10.3390/microorganisms7110534_

Round 1

Reviewer 1 Report

In the present study the authors examined the virulence properties of Escherichia coli ST131 containing  bacteriocin-encoding plasmids. The findings are of interest, demonstrating that plasmids harboring bacteriocins give additional advantages for highly virulent and resistant ST131 isolates, improving the ability of these isolates to compete with other microbiota for a niche and thereby increasing the risk of infection. Additionally, the study is well organized and well presented.

Comments:

Line 59-60: add 'and carbapenems'.

Line 359: isolates 5848 and 5332 are clinical strains, while E. coli DH5α is a laboratory strain. Thus, it

would be more correct to transfer bacteriocin-encoding plasmds to E. coli laboratory strain or to use as  negative control clinical strains not-producing bacteriocins, and to repeat the experiment.

Lines 460-463: this finding could be associated with increased copy numbers (and thus expression) of ColE plasmid, while the plasmids characterized from clinical strains are low copy. This part should be reformatted. 

Author Response

Dear reviewer we would like to thank you for reviewing our manuscript. Please find below a point-by-point response to your comments.

Line 59-60: add 'and carbapenems'

Thank you for the suggestion. We have included it in the corrected version of the manuscript.

Line 359: isolates 5848 and 5332 are clinical strains, while E. coli DH5α is a laboratory strain. Thus, it would be more correct to transfer bacteriocin-encoding plasmids to E. coli laboratory strain or to use as  negative control clinical strains not-producing bacteriocins, and to repeat the experiment.

We apologize for not being clear and we are not sure if we fully understand your comment. In our experiments we used the DH5-alfa laboratory strain as a negative control as it is often used in other studies as well when doing these kind of experiments. Indeed, as you suggested we also transferred a bacteriocin-encoding plasmid to the laboratory strain for comparison with the wild-type strain. In addition, both bacteriocin-producing and non-producing clinical isolates were compared to each other. From the non-bacteriocin producing isolates only isolate 1643 showed a statistically significant lower adherence compared to isolate 5332 (p=0.0429) and no other statistically significant differences were observed. 

To make it more clear we have changed the sentence into:

"As expected, our results showed that the bacteriocin-producing isolates 5848, 5332 and the DH5α+ (transformant) had a higher adherence to HEK-293 cells, compared to the negative control (DH5α)."

Lines 460-463: this finding could be associated with increased copy numbers (and thus expression) of ColE plasmid, while the plasmids characterized from clinical strains are low copy. This part should be reformatted. 

We thank the reviewer for this suggestion we included a remark on this in the discussion by adding the following sentence.

"In addition, a higher copy number of the plasmid in the transformant strain compared to that of the plasmid in clinical isolates, and thus a higher expression of the bacteriocin gene located on it, may be associated with an increase in virulence. However, no difference in virulence was observed between the transformant strain and the clinical isolates."

Reviewer 2 Report

The authors aimed to investigate the bacteriocin activity in-vivo in E.coli isolates  especially in ST131 isolates due to its clinical impact on public health. The study indeed is original and little has been reported on this aspect in the literature. The author presented the different aspects needed for the assessment of the aim of the study and the outcome supported their hypothesis. The authors also addressed almost all the variables needed to be addressed.

Only few minor comments to address:

line-122-134: the contigs used for plasmid identification ;which assembly was used? is it the one from the illumina assembly or the hybrid? It would be useful to present the results as i expect to have at least closed plasmids using the hybrid assembly especially that the sizes mentioned could be closed by the technique used here. if this is the case then why would you need to blast the contig against closest reference plasmids?1 this point is not entirely clear. And in the case that the plasmid contigs are broken to more than one contig, in-silico closure of the plasmids are not enough and should be confirmed through PCR gap closing technique and size confirmation using S1 PFGE.

line-385-389: the AmpC-β-lactamase production was found on the chromosome of the genome or plasmid?! The plasmid encoded resistance could not be only shared because of the probable cause of a lineage evolving from another. these could be shared as well through conjugation especially that these isolates are from the same geographic are.

line 420-421: This point is very important. were you able to find any other genes in ST69 and ST405 that might cause this observed resistance to bacteriocins. if not this could be an important thing to follow up in future.

Author Response

Dear reviewer we would like to thank you for reviewing our manuscript. Please find below a point-by-point response to your comments.

line-122-134: the contigs used for plasmid identification ;which assembly was used? is it the one from the illumina assembly or the hybrid? It would be useful to present the results as i expect to have at least closed plasmids using the hybrid assembly especially that the sizes mentioned could be closed by the technique used here. if this is the case then why would you need to blast the contig against closest reference plasmids?1 this point is not entirely clear. And in the case that the plasmid contigs are broken to more than one contig, in-silico closure of the plasmids are not enough and should be confirmed through PCR gap closing technique and size confirmation using S1 PFGE.

We apologize for not being clear. We used Unicycler for constructing hybrid assemblies to analyze the plasmid structure. In the last step Unicycler does a circularization step at the end of the pipeline where it looks for overlapping regions to close the plasmids/chromosome. In addition, we blasted our plasmids (existing of only one contig) to check if it was similar to other plasmids, especially to those that are common among foodborne pathogens present in the gut. 

To clarify this we have adapted the method section "Plasmid analysis and identification of bacteriocin genes". It now reads:

"The plasmids incompatibility groups were identified by uploading the assembled files, generated using the hybrid assembly approach described above, to PlasmidFinder (version 1.3) [26]. The plasmid sequences were annotated automatically using the RAST server version 2.0 and manually using CLC Genomics Workbench v12.0 (Qiagen, CLC bio A/S, Aarhus, Denmark). Subsequently, plasmids were uploaded to BLAST (NCBI database) to identify the closest reference plasmids. Alignment of plasmid sequences was done using Easyfig v2.2.3 [30] and DNA plotter [31]. The bacteriocin genes present in the isolates were detected by BAGEL 3, by uploading the fasta files onto the online tool [32]. Plasmid mobility was analyzed by locating and typing the origin-of-transfer (oriT) regions using a DNA structural alignment algorithm that finds minimal Euclidean distances and p-values between query oriTs and target plasmids [33]. Potential transfer host ranges of the predicted MOB groups were determined from a MOB-typed dataset [34]."

In addition, we adapted figure 3, using DNA-plotter instead of CGEview, to make clear that the plasmids are closed and not broken. 

line-385-389: the AmpC-β-lactamase production was found on the chromosome of the genome or plasmid?! The plasmid encoded resistance could not be only shared because of the probable cause of a lineage evolving from another. these could be shared as well through conjugation especially that these isolates are from the same geographic are.

We thank the reviewer for this comment. Indeed the AmpC-β-lactamase gene is located in the plasmid and therefore of plasmid transfer between isolates collected in the same geographic area could have occurred. However, many ST131 isolates present in the same geographic area did not contain the plasmid. Since we can not exclude it did happen for the bacteriocin-producing ST131 isolates, we changes this part of the discussion as follows:

"Horizontal transfer of the plasmid between isolates within the same geographic area could have occurred. However, also many ST131 isolates were circulating in the same area without these plasmids. Therefore, the presence of similar plasmids in both resistant H22-ST131 and susceptible H30-ST131 isolates could also be explained by the fact that H30 strains evolved from the H22 lineage [21]."

line 420-421: This point is very important. were you able to find any other genes in ST69 and ST405 that might cause this observed resistance to bacteriocins. if not this could be an important thing to follow up in future.

We thank the reviewer for this suggestion. We did not find any gene that we can relate to the resistance to bacteriocins identified in this study in the ST69 and ST405 isolates. However, we included in the discussion a phrase, saying that it is a topic for future investigation:

"Interestingly, despite the resistance of ST69 and ST405 to bacteriocins, we did not identify any bacteriocin genes nor did we find any other known genes related to bacteriocin resistance in these groups. Therefore, it would be highly interesting to investigate the mechanism(s) behind the observed resistance in future research."

Round 2

Reviewer 1 Report

The authors have successfully adrressed the points raised. Additionally, tha manuscript has been significantly improved.